# Validation of Two Instruments for the Correct Allocation of School Furniture in Secondary Schools to Prevent Back Pain

**DOI:** 10.3390/ijerph19010020

**Published:** 2021-12-21

**Authors:** Alfonso Gutiérrez-Santiago, Iván Prieto-Lage, José María Cancela-Carral, Adrián Paramés-González

**Affiliations:** 1Observational Research Group, Faculty of Education and Sport, University of Vigo, 36005 Pontevedra, Spain; ags@uvigo.es (A.G.-S.); aparames@uvigo.es (A.P.-G.); 2HealthyFit Group, Faculty of Education and Sport, University of Vigo, 36005 Pontevedra, Spain; chemacc@uvigo.es

**Keywords:** school furniture, education, mismatch measures, validation, back health

## Abstract

Background: Back pain is common in secondary school students. If we adjust the school furniture to the anthropometric characteristics of the pupils, we will improve their posture and reduce back pain. There is a high degree of mismatch between the furniture used by students and that which should be used. The objectives of this research are to discover the degree of mismatch and validate two instruments that allow a correct allocation of the furniture. Methods: The selected sample was 132 secondary students (14.08 ± 1.10 years). An anthropometer was used to determine the ideal height of the chair and table; data were taken from body segments. The recorded values were compared with those obtained by the two measurement instruments to be validated. Inter-measurer and intra-measurer reliability and validation were performed using t-tests and Pearson’s coefficient, respectively. Different analysis techniques were used: descriptive, one-way ANOVA, t-test, and effect size. The established level of significance was ρ < 0.05. Results: The mismatch between the anthropometric dimensions of the students and the existing furniture in the classrooms was 98.5 % for the chairs and 100 % for the tables. The correlational analysis of the instruments to be validated shows an r = 0.993 in the chair and r = 0.996 in the table. Conclusions: There is a high degree of mismatch between the furniture and the anthropometric characteristics of the students. The proposed furniture allocation instruments are adequate.

## 1. Introduction

Neck and back pain are the fourth most common causes of disabilities worldwide [1]. The presence of this pathology during adolescence increases the risk of chronicity in adulthood [2]. In this regard, back pain is very common in secondary education [3].

Multiple factors may influence back pain in schoolchildren [3]. Most of the studies that were analysed in a systematic review [4] observed that an adequate adjustment between the dimensions of the furniture and the anthropometric characteristics of the students had an impact on improving their posture and reducing pain.

Students spend a quarter of the day in school, and 80% of that time is spent engaging in various activities related to reading and writing, which require them to be seated for many hours [5,6]. Therefore, it is essential that school furniture adapts its anthropometric characteristics; otherwise, it is possible that anatomical-functional disorders and problems regarding the learning process will develop [7,8].

A longitudinal study of growth in Spain [9] indicates that our test students are in a stage where each student has their own maturation *tempus*, with different growth rates. Before puberty, legs grow faster than the trunk of the body, and in teenagers, growth is mainly in the trunk [10]. Thus, at these ages, high growth stages can occur and therefore increase the amount of postural and musculoskeletal dysfunctions [11]. There are publications that show high levels of musculoskeletal pain in similar populations [12,13], which can lead to the appearance of future injuries [14]. Numerous studies indicate that the pain observed in teenagers can be attributed to several factors, one of the most relevant being the mismatch between the anthropometric characteristics of the students and the dimensions of the furniture [6,7,15]. On the other hand, we know that the use of suitable furniture can lead to a reduction in fatigue and discomfort in a seated posture [4,14]. Therefore, it is necessary to develop an intervention that minimises mismatches. Because of this, we must consider the various growth stages and maturation of the students at these ages. 

European schools should follow the standard EN 1729-1 [16] that determines the dimensions of the classroom furniture, which proposes eight sizes applicable to the dimensions of desks and chairs to cover the variability associated with the students’ anthropometry. In Spain, there is only one investigation on anthropometry and school furniture [17], in which some grades are analysed, but without the instruments or measurements that are necessary to determine the ideal chair and desk. It is necessary to carry out anthropometric studies to analyse the degree of adjustment and check if the EU catalogue [16] is suitable for students. Another aspect to consider that justifies the need for the study is that the European reference is created from anthropometric data from students in the United Kingdom. Spanish growth studies indicate that changes in height have been taking place in recent decades despite regional differences, with measurements becoming more similar to those from other European countries, but remaining below some central and European nations [18]. 

Scientific literature has shown that, regardless of whether the furniture follows the sizes of the established catalogue, there is a high degree of mismatch between the seats and desks that students use and those they should use due to their anthropometric characteristics. There are works about it in the United States [14,19], South America [11,20], Africa [21], Asia [22], Oceania [23], and in Europe [12,24,25,26]. In all these studies, they conclude there is a mismatch. They usually indicate the sizes that should be used, but few studies provide solutions. Some studies are in favour of proposing more than one desk and chair size per academic year [12,14,24,25], proposing, in turn, that the furniture be assigned based on anthropometric dimensions and not by chronological age [19,21,27], due to the high variability of the anthropometric dimensions among students of the same age. 

Schools that follow the regulations and have different furniture sizes require that their teachers know how to associate this furniture with morphological characteristics since ignorance can lead to mismatch [28]. The selection of school furniture for students by their teachers is a complex task [25]. European regulations specify that instructions should be given on how to adjust the furniture [16], but do not make any specific proposal beyond making a recommendation for ranges based on height or popliteal height. The experts indicate that the popliteal height is the most accurate measurement to determine the ideal chair size [29], demonstrating, in turn, that there is no correlation between this measurement and height. Accurate popliteal measurement requires experience and skill [30]. In addition, a specific measuring instrument such as the anthropometer is needed, and a series of measurements are needed to apply the formulas for calculating the ideal furniture [31]. This entire procedure is unaffordable for teachers, making it necessary to look for simple measurement strategies that are easy to apply [28,32] and allow the allocation of adequate furniture in schools. 

Therefore, the objectives of this research are to analyse the degree of adjustment of furniture in relation to the anthropometric characteristics of secondary schools’ students, check if the size system of the European Union (EU) catalogue is adjusted to the population under study, and validate a system that allows teachers to easily and adequately assign school furniture.

## 2. Materials and Methods

### 2.1. Sample 

Based on the Spanish scholar system, secondary school has 4 grades, with students ranging from 12 to 16 years old. The participants in this study were students from the 1st to the 4th of secondary school from the north of Spain (grades 7–10 United States, Years 8–11 United Kingdom). A total of 132 students (55 boys and 77 girls, average age 14.08 ± 1.10 years; [males - 13.99 ± 1.09 years; females - 14.14 ± 1.11 years]) participated in the study. 

Authorisation permission to carry out data collection tasks at the school was requested by the centre’s management. All families and students were informed about the study’s objectives and read and signed an informed consent form. The ethical principles of medical research on human beings of the Declaration of Helsinki were respected [33]. The study was approved by the ethical committee of the Faculty of Education and Sports Sciences of the University of Vigo with the code 04/1019.

### 2.2. Procedure

#### 2.2.1. Anthropometric Characteristics and Furniture

For the calculation of the ideal seat height (SH) and ideal desk height (DH) for each student, anthropometric data were taken and two formulas were applied [34].

For all measurements, an anthropometer was used (Cescorf 60 cm, precision 0.01 cm, Porto Alegre, Brazil), approved by the International Society for the Advancement of Kinanthropometry (ISAK). Height was measured with a measuring rod (Seca portable stadiometer 20–205 cm, Hamburg, Germany), and a scale was used for weight measurement (Tanita UM-076, precision 2g, Tokyo, Japan).

Each parameter was measured at least twice, and if values showed a difference of more than 0.5 cm, an additional measurement was taken. The following anthropometric measures were considered to estimate ideal furniture dimensions: [35].

Height: vertical distance from the floor to the top of the head, with the subject standing upright and looking straight ahead (Frankfurt plane).Shoulder height sitting (SHS): vertical distance from the subject’s seated surface to the acromion.Elbow height sitting (EHS): measured with the elbow flexed at 90°. It is the vertical distance from the tip of the elbow (olecranon) to the subject’s seated surface.Popliteal height (PH): measured with the knees flexed at 90°. This is the vertical distance from the floor to the posterior surface of the knee (popliteal surface).

Anthropometric measurements were taken by following the procedure established in other similar studies [4]. For the assessment, students’ measurements were taken on the right side (except height and weight), with the participant sitting on an adjustable-height chair with a horizontal surface seat, with the legs flexed at 90° and the feet resting on an adjustable footrest. During the measurement process, the participant was barefoot and wearing trousers and a T-shirt.

The anthropometric data obtained were compared with the dimensions of the furniture (Class Seat Height and Class Desk Height) to identify a match or a mismatch between them, defining a mismatch as the lack of coincidence between the dimensions of the furniture and the anthropometry of the students, according to the following formulas [34]:Seat Height (SH): (PH + 2.5) cos30° ≤ SH ≤ (PH + 2.5) cos5°Desk Height (DH): (SH + EHS ≤ DH ≤ (SH + EHS × 0.7396 + SHS × 0.2604)

The evaluated students always used the same classroom and the same desk and seat. The sizes of the furniture in this school were all the same, with the models of 48 cm seats and 78 cm desks from the regional education administration [36], different sizes from those contemplated by European regulations [37]. In any case, the seat and desk associated with each student were measured for possible wear and tear, given the age of the furniture. In this way, the dimensions of the furniture that were collected were: Class Seat Height (CSH): the vertical distance from the ground to the midpoint of the extended edge of the seat surface.Class Desk Height (CDH): the vertical distance from the floor to the top of the front edge of the desk.

#### 2.2.2. Size System of the EU and Galician Catalogue

Once the ideal seat and desk heights (SH and DH) were determined for each student, the existence of a size that adjusted to their anthropometric dimensions according to the EU catalogue [16] and the region of Galicia [36] was tested. 

The EU catalogue does not determine the sizes by school stage but by popliteal height or stature, establishing eight possible sizes (21, 26, 31, 35, 38, 43, 46, and 51 cm in seats and 40, 46, 53, 59, 64, 71, 76, and 82 cm in desks). 

The catalogue from the Galician region determines four sizes (36, 40, 44, 48 cm in seats, and 60, 66, 72, 78 cm in desks) for primary education (6–12 years) and secondary education (13–18 years). 

#### 2.2.3. Ideal Seat Height Test (ISHT) and Ideal Desk Height Test (IDHT)

The Ideal Seat Height Test (ISHT) and the Ideal Desk Height Test (IDHT) are 3-millimetre PVC vinyl templates (see Figure 1). Both tools have been designed following the eight furniture sizes proposed by the EU catalogue [16]. Each colour determines the measurement of the ideal size in centimetres and millimetres. The ranges established between each of the bands were extracted from the equations of previous researches [34]. These measurement instruments can be downloaded for free in .tiff format from the research group website: https://www.iobserving.com/p/mobiliario-furniture.html (accessed on 1 December 2021). Afterward, they can be printed on a special printer. 

The research has taken account of the ergonomic recommendations which specify that, when seated, the soles of the feet should be in contact with the floor and the knees at right angles to the floor. The appropriate indicator for correct chair adjustment is the popliteal height [19]. 

Our design of the ISHT instrument is based on the Peter lower leg meter [28], a non-validated instrument that uses a strategy that facilitates the measurement of popliteal height using a system of colours whose objective is determining the seat height. 

In our study, we placed the instrument on a desk. Dropping the part with coloured stripes of the ISTH vertically, so that when students sat at the desk (on the ISTH) and placed the popliteal space on the edge of the desk, the height at which the sole would reach when barefoot and with the heels at right angles to the instrument, would indicate the ideal height for the chair (see Figure 2 left).

The IDHT was created based on ergonometric guidelines, which indicate that the desk height should be at the elbow height [7,19], considering that the trunk should be upright, the arm vertical and the forearm horizontal, forming a right angle at the elbow [20]. 

The IDHT was placed against the wall. The students were then seated (with shoes) on the chair previously adjusted to the ISHT. They were positioned next to the instrument so that, by forming a right angle at the elbow, the lower part of the forearm indicated the ideal height of the desk (see Figure 2 right). No other similar instrument has been found in literature.

#### 2.2.4. Data Collection 

Before starting the study, training sessions were held for two weeks (one hour a day) to measure the anthropometric parameters to be taken later, in order to reduce the differences between the two measurements. At the end of the training sessions, the intra- and inter-measurement reliability were evaluated. 

Ten days passed during the data recording process, two days to obtain the data used to analyse the degree of intra-measurer and inter-measurer reliability, four days to record the anthropometric data using an anthropometer as a measuring instrument, and another four days for measurements with the two instruments to be validated: the ISHT and IDHT. The sessions were held during school hours from 9:00 to 14:30 during the month of November. 

To carry out the measurement process, two work teams were created using the recommendations of previous scientific studies as references [38]. Each team consisted of four people: a measurer, a data logger, a sample organiser, and another person to support the measurer. The inter-measurer and intra-measurer reliability tests were carried out with a heterogeneous group based on the grade and gender of the 25 students [39]. In the first class of the day, the two measurers took data of the body dimensions with the anthropometer, and at the end of the morning the measurers used the ISHT and IDHT with these same participants. The following day, the same procedure was applied, obtaining a value higher than 0.95 in both the inter-measurer and intra-measurer tests. All measurements were carried out by two specialist anthropometrists with ISAK 3 level (International Society for the Advancement of Kinanthropometry) and with previous experience in this type of assessment. A minimum of two measurements were taken for each parameter. If the values found varied more than 0.5 cm between them, an additional measurement was performed [12].

### 2.3. Statistical Analysis

All statistical analyses were performed using IBM- Statistical Package for the Social Sciences, version 20.0 (IBM-SPSS Inc., Chicago, IL, USA). The Kolmogorov-Smirnov test confirmed the normality of the sample. A descriptive analysis, stratified by grade, of each of the variables under study was carried out through measures of central tendency (mean and standard deviation). The mean values of the parameters obtained in the different grades were compared using one-way ANOVA, applying a Turkey-b post hoc test in the case of statistically significant differences (*p* < 0.05). The aforementioned mean values were also compared between men and women, using a *t*-test for independent samples. A comparison of means was conducted using the *t*-test for related samples to observe the differences between the real values (furniture they had in the classroom) and the ideal values (calculated with the two analysis methods –traditional and new). Additionally, the effect size was analysed using Cohen’s d (d < 0.2 -null-, d = 0.2–0.49 -small-, d = 0.5–0.80 -moderate- and d > 0.8 -large). Reliability tests were performed using paired samples *t*-tests, with a 95% confidence interval. Pearson’s correlation coefficient was used for the validation of the ISHT and IDHT (0.90 to 1.00 very high; 0.70 to 0.89 high; 0.50 to 0.69 moderate; 0.30 to 0.49 low; 0.00 to 0.29 negligible [40]). The Bland and Altman analysis [41] and its corresponding linear regression test were also performed. In all statistical tests, the level of significance was *p* < 0.05. 

## 3. Results

Table 1 shows the anthropometric characteristics of the analysed students. 

The results show the existence of significant differences in the anthropometric records made depending on the grade to which the tested students belonged (*p* < 0.05) for all the variables analysed. Intergroup analysis showed that the most notable differences were found between the 1st secondary school grade and the other three grades. 

Significant differences were observed based on gender analysing the group globally in the variable height (t = 4.599; sig = 0.001). The analysis of the size effect (Cohen’s d) indicated that the found differences were moderate (d = 0.79). When performing this test classifying the data by grade, such significant differences were not evidenced. 

Table 2 and Table 3 show the size of the chair and desk used by the students before the investigation (real height in class), as well as the one estimated later as being ideal with its minimum and maximum interval recommended both with the traditional instrument (anthropometer) and with the new measurement tool (ISHT and IDHT).

Table 4 shows the furniture assignment that should be used by grade, according to the EU and Galician region catalogues. 

The results obtained showed large mismatches between the anthropometric characteristics of the students and the height of the furniture they use in their classrooms. Thus, the frequency analysis showed that 98.5% of the participants in this research used a seat that was not consistent with their anthropometric dimensions, and 100% used a desk that was not appropriate. In general, the students used a chair between eight and 10 cm higher than what corresponded to them, and used a desk between 12 and 16 cm larger than recommended. In addition, whether the students had the same size of seat and desk was verified (same colour), and a mismatch of 49% was observed.

Regarding the ideal seat and desk measurements (both using the anthropometer and the new proposal), statistically significant differences were found between the different grades (*p* < 0.05). As in the previous case, the most relevant intergroup differences occur between the 1st year of secondary school students compared to the other grades.

The comparative analysis of the data recorded according to gender (without segmenting by grade) revealed that there were statistically significant differences (*p* < 0.05) in the ideal variable desk height with the anthropometer (t = 2.604; sig. = 0.080; d = 0.45) and IDHT (t = 2.564; sig. = 0.011; d = 0.45). The analysis of the size of effect (Cohen’s d) shows that the differences found in both cases were small. However, in the comparative analysis by gender segmented by grade, such differences ceased to exist. 

The T analysis for samples related to the real height between the furniture and the ideal height after the anthropometric analysis showed statistically significant differences (*p* < 0.001) in all the comparisons made. The effect size analysis (Cohen’s d) indicated that all the differences found were large (d > 0.8).

Results of the furniture assignment according to the catalogues that were analysed (Galician region and EU) determined they would need between two and four different sizes per grade (in both cases) to provide the students with what they need. The use of the proposed sizes in both catalogues is appropriate. 

The correlational analysis, shown in Figure 2, between the new measurement tool to calculate ideal size for the seat (ISHT) and the desk (IDHT) compared to the traditional one (using the anthropometer) showed a high correlation in both cases (r = 0.993 in chairs and r = 0.996 in desks). Analysing the group segmented by grade and sex, the degree of correlation continued to be high (ISHT 1st to 4th show r > 0.992; ISHT males and females showed r > 0.993; IDHT 1st to 4th show r > 0.993 and IDHT males and females show r > 0.996) (Figure 3). 

Considering the statistical methods for assessing agreement between two clinical Bland and Altman [41] measurement methods, the following plottings have been established. In the linear regression test, no statistically significant differences were observed between the two methods, neither in the assessment of the ideal seat height (0.182) nor in the ideal desk height (0.765) (Figure 4).

## 4. Discussion

The purpose of this research was to analyse the degree of mismatch between the furniture and the anthropometric characteristics of the students, to contrast whether the sizes of the European regulations were adapted to the population group, and to validate a new simple system that would allow a more suitable distribution of chairs and desks. 

The population under study showed a degree of maturation typical of the various growth stages at these ages [9], with gender having a small effect, while greater differences existed between the 1st year of secondary school students and the other three grades.

The secondary school students (12–16 years) who participated in the research all had a single size of furniture. Based on the growth data concerning this stage, it was expected that a single desk and seat size would not allow adaption to their characteristics [21]. This circumstance led to an extremely high degree of mismatch of greater than 98%. Although there is no similar research data in Spain, these results align with other research carried out among the European population. In a study in the north of Portugal [12], a 96% mismatch percentage was obtained for seats and 76% for desks. Although they obtained a degree of concordance in the desk that was higher than that of the present intervention, the adjustment percentages are also very low. In Greece [24], they determined that the height of the chair and the desk were higher than the ideal limits accepted for most of the students, with a slightly lower mismatch: 71.5% and 81.8%, respectively. 

The EN 1729-1 guide [16] for the correct adjustment of school furniture was introduced in Europe in 2004. However, the existence of these regulations has not brought about a decrease in the degree of mismatch in the classrooms. This problem is not unique to this continent as it also occurs in populations worldwide, although experts advise caution when making anthropometric comparisons between populations from different geographical areas [42]. For example, in Chile [11], the size of the chair is higher than the ideal size in 81.33% of the cases, with the desks being higher than the recommended size in 100% of the cases, values quite similar to those found in the reference study. In North America, students were also using an inadequate seat in 92% of the cases, and an incorrect desk 95.1% of the time [14]. Similarly, in Saudi Arabia [43], a mismatched chair height of 80% and a mismatch of desks of 79.2% was found. 

This generalised mismatch displays a reality that scientific literature has already pointed out. Despite the fact that the administrations promote regulations that attempt to control and standardise the size of furniture in different countries, the allocation of equipment depends on the educational centre and is done without taking the anthropometric characteristics of the students [27] into account. 

The EU catalogue [16] proposes up to eight sizes regarding furniture allocation for students (from 21–51 cm for chairs and 40–82 cm for desks). This catalogue would allow an adequate allocation based on the determined values of the ideal chair and desk height evidenced in this study, and it appears the size proposal is valid. However, this would also occur (at least in secondary school) if the furniture were assigned according to the current regional catalogue [36], which presents ranges of from 32–48 cm in chairs and from 48–78 in desks, so it appears that the problem is not so much the sizing proposal but the allocation carried out later in schools. The solution, therefore, would not be determined by acquiring or equipping the centres with new furniture sized according to the EU catalogue, since if it is not correctly assigned the problem will persist. This has already been investigated in Portugal [12], where the school, with new furniture corresponding to EU sizes, carried out a reallocation that barely improved the previously existing mismatch. In this study, as happened in a similar study of primary education [32], between two and four different sizes were needed per grade. In any case, all secondary education (12–16 years) would have four seat sizes and three desk sizes both in the regional [36] and the EU catalogues [16]. Tests show that no student is excluded from the furniture assignment using these catalogues, which is something that had occurred with the regional catalogue in primary education [32].

Currently, there is one seat and desk size per grade in most schools. The high standard deviation found in the different anthropometric measurements implies having students from the same grade in different measurement percentiles. It seems logical that the furniture allocation system is not guided by chronological age, requiring several seat and desk sizes per academic year. This has been recognised on numerous occasions by the scientific community, and we agree with the collected data [12,14,23,24,25], along with the use of adjustable furniture [22,43,44,45,46,47]. 

For the viability of its application in the classroom, and as recommended by experts, the procedure should depend on anthropometry. However, it requires expensive measurement instruments, training, and anthropometric knowledge, skills which are non-existent for most teachers [31]. In turn, a system based on anthropometric measurements would lead teachers to invest a lot of time in this process, so it is necessary to develop a simple and easy-to-use system for teachers, allowing easy allocation of furniture in the classroom as other researchers propose [28]. In this regard, the European legislation [16] proposes to perform this task based on height or popliteal height. There is no doubt that the simplest and most objective indicator those teachers would have to carry out the assignment would be the height. But, as other authors state [29], we agree that its use is incorrect since we have corroborated the low efficiency of this allocation method (based on the EU catalogue sizes, errors of 92.4% were found in seats and 100% in desks). This problem could be solved using tools such as ISHT and IDHT.

In the absence of specific instructions from the EU catalogue [16] to enforce its rule, other proposals have been produced. In one of them, the researchers [23] suggest that students follow a guide and, after trying different sizes, select the seat based on their self-perception of comfort. This system based on the subjective perception of the students does not meet the precise validity and reliability criteria [48]. Another mechanism is the “Peter lower leg meter” [28], which measures the popliteal height. This method is recommended by other authors [29,46]; it is unknown whether or not it follows a validation system. 

The aforementioned solutions and the EU catalogue [16] incepted from the idea that when defining the popliteal height and estimating chair size, the size of the desk can be associated. From the data obtained, this would be an error, as in almost half the cases, the students needed different sizes of chairs and desks. This may be because in certain growth stages, the growth is not simultaneous and proportional between the torso and the extremities of the body [10], as it is possible for some students to have similar popliteal heights but also have different elbow and shoulder heights. Therefore, the size should not be assigned as a set from the popliteal height, thus requiring different combinations of seat and desk size [22]. 

The two tools validated in this research can contribute to solving the existing problem regarding the allocation of furniture in schools, providing a size related to colour according to the EU catalogue, applicable to both fixed and adjustable furniture. It is also proposed that the schools take inventory with coloured stickers to catalogue the available furniture, which would make it easier for students to use the appropriate furniture when making classroom changes. It has been shown that the proposed method is a system with a high correlation to the validation process regarding the anthropometer reference method. At the same time, it is a simple tool that is available to any educational professional with some basic introductions and, finally, it takes into account the choice of the ideal chair and desk height independently.

### Limitations of the Study and Future Perspectives

The results discovered refer to a small sample that should be expanded in future to confirm the legitimacy of the validation found. Likewise, it would be interesting to extend the validation process of ISHT and IDHT to other student populations, as in other scholarly stages, such as in pre-school, primary, secondary/high school, or adult training. On the other hand, it would be interesting to contrast the reliability of secondary school teachers using the ISHT and IDHT themselves, since in this study the tool has been used by expert and trained researchers. A tool can be valid, but it also needs to be reliable.

## 5. Conclusions

There is a high degree of mismatch between the furniture and the anthropometric characteristics of the students studied. It has been proven that the catalogue of sizes from the EU and the regional educational administrations would cover the students’ needs if they were correctly assigned in the classroom. It has been confirmed that a unique chair and desk size for each grade cannot be established, given that there is a high variability of anthropometric measurements in each age group, which requires two or three different sizes in each grade or adjustable furniture.

The high correlation found when comparing the ideal height determined with ISHT and IDHT, with respect to the anthropometric estimate, provides schools with a new simple tool to assign furniture appropriately. 

## Figures and Tables

**Figure 1 ijerph-19-00020-f001:**
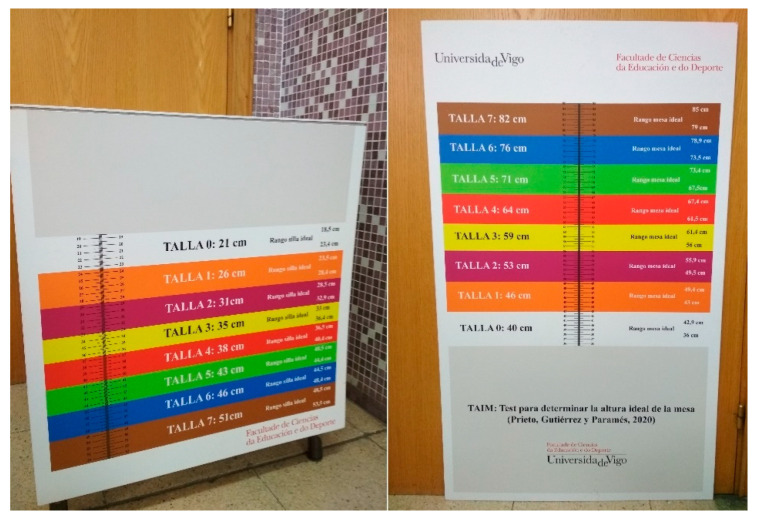
Measuring tool ISHT and IDHT.

**Figure 2 ijerph-19-00020-f002:**
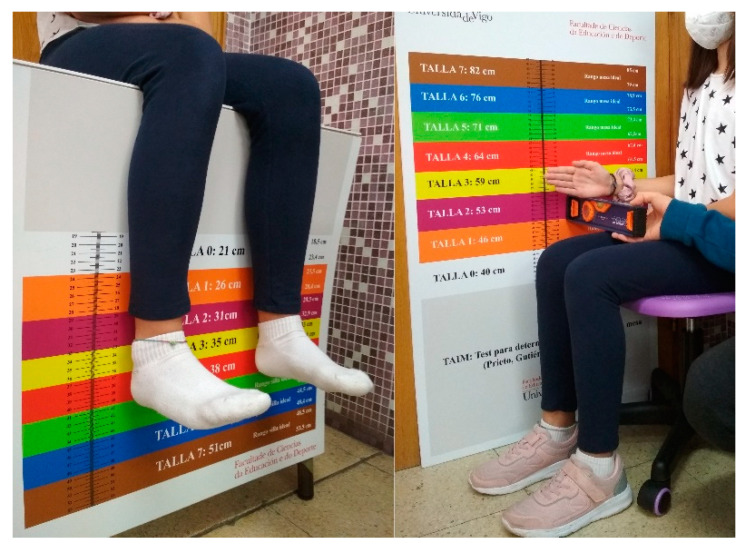
Use of measuring tool ISHT and IDHT.

**Figure 3 ijerph-19-00020-f003:**
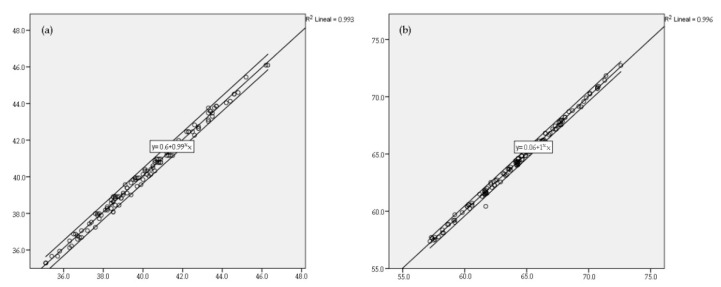
Correlational analysis between the ideal height of the chair calculated with an anthropometer and the ideal height of the chair with ISHT (**a**), and correlational analysis of the ideal height of the desk calculated using the anthropometer and ideal height of the desk using IDHT (**b**).

**Figure 4 ijerph-19-00020-f004:**
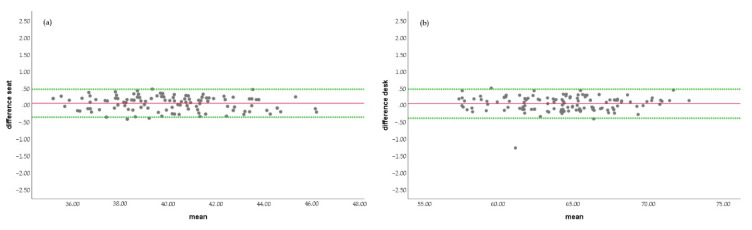
Bland and Altman test for ISHT (**a**) and IDHT (**b**).

**Table 1 ijerph-19-00020-t001:** Descriptive analysis and ANOVA test of the anthropometric values of the students by grade.

Grade	Age Range (years)	Height (cm)	Weight (kg)	Popliteal Height (cm)	Elbow Height (cm)	Shoulder Height (cm)
1st grade (*n* = 32)	12–13	151.10 ± 6.64 ^a^	46.73 ± 10.99 ^a^	35.88 ± 1.93 ^a^	16.43 ± 2.53 ^b^	49.13 ± 3.06 ^a^
2nd grade (*n* = 32)	13–14	160.16 ± 6.36	55.90 ± 10.37	37.81 ± 2.08	17,10 ± 1.91 ^b,c^	51.77 ± 2.65 ^d^
3rd grade (*n* = 42)	14–15	163.26 ± 9.19	61.13 ± 13.81	37.84 ± 3.06	17.57 ± 2.59 ^b,c^	52.96 ± 3.20 ^d,e^
4th grade (*n* = 26)	15–16	162.81 ± 7.73	60.03 ± 12.9	37.53 ± 2.70	18.30 ± 2.95 ^c^	53.67 ± 2.92 ^e^
Anova	F	16.744	9.562	4.540	2.912	14.089
	g/l	3	3	3	3	3
	Sig.	0.001	0.001	0.005	0.037	0.001

^a^ 1st shows differences with the others; ^b^ 1st to 3rd with 4th; ^c^ 2nd to 4th with 1st; ^d^ 2nd to 3rd with the others; ^e^ 3rd and 4th with the others.

**Table 2 ijerph-19-00020-t002:** Comparison of chairs between grades (Anova) of the ideal height/range with the anthropometer and the ideal height using ISHT. Comparison by grade of the real height with the ideal height with the anthropometer and the real height with the ideal height using ISHT (*t*-test for related samples).

Grade	Age Range (years)	Real Height in the Classroom (cm)	Ideal Height/Range with the Anthropometer (cm)	Ideal Height with ISHT (cm)	*t*-Test Real—Ideal Height with Anthropometer		Cohen’s d		*t*-Test Real—Ideal Height with ISHT		Cohen’s d	
					t	*p*	d	r	t	*p*	d	r
1st grade (*n* = 32)	12–13	47.81 ± 0.18	38.81 ± 1.80 ^a^ (36.10 ± 1.68/41.49 ± 1.91)	38.70 ± 1.80 ^a^	28.871	0.001	7.04	0.96	29.213	0.001	7.12	0.96
2nd grade (*n* = 32)	13–14	47.87 ± 0.16	40.61 ± 1.94 (37.78 ± 1.81/43.41 ± 2.08)	40.60 ± 2.01	21.555	0.001	5.27	0.94	20.804	0.001	5.10	0.93
3rd grade (*n* = 42)	14–15	47.91 ± 0.13	40.63 ± 2.85 (37.79 ± 2.65/43.45 ± 3.03)	40.59 ± 2.89	16.766	0.001	3.61	0.87	16.581	0.001	3.58	0.87
4th grade (*n* = 26)	15–16	47.86 ± 0.27	40.35 ± 2.51 (37.52 ± 2.35/43.15 ± 2.69)	40.29 ± 2.43	15.492	0.001	4.27	0.91	16.177	0.001	4.45	0.91
Anova	F	1.564	4.54	4.85	-	-	-	-	-	-	-	-
	g/l	3	3	3	-	-	-	-	-	-	-	-
	Sig.	0.201	0.005	0.003	-	-	-	-	-	-	-	-

^a^ 1st shows differences with the others.

**Table 3 ijerph-19-00020-t003:** Comparison of desks between grades (Anova) of the ideal height/range with the anthropometer and the ideal height using IDHT. Comparison by grade of the real height with the ideal height with the anthropometer and the real height with the ideal height using IDHT (*t*-test for related samples).

Grade	Age Range (years)	Real Height in the Classroom (cm)	Ideal Height/Range with the Anthropometer (cm)	Ideal Height with IDHT (cm)	*t*-Test Real—Ideal Height with Anthropometer		Cohen’s d		*t*-Test Real—Ideal Height with IDHT		Cohen’s d	
					t	*p*	d	r	t	*p*	d	r
1st grade (*n* = 32)	12–13	77.88 ± 0.13	61.83 ± 3.28 ^a^ (57.74 ± 3.18/65.92 ± 3.40)	61.81 ^a^ ± 3.31	27.918	0.001	6.90	0.96	27.671	0.001	6.84	0.96
2nd grade (*n* = 32)	13–14	77.86 ± 0.13	64.56,4 ± 2.62 (60.21 ± 2.49/68.92 ± 2.79)	64.59 ±2.58	29.007	0.001	7.17	0.96	29.453	0.001	7.26	0.96
3rd grade (*n* = 42)	14–15	77.90 ± 0.16	65.15 ± 3.46 (60.70 ± 3.25/69.59 ± 3.68)	65.08 ±3.46	24.098	0.001	5.21	0.93	24.223	0.001	5.23	0.93
4th grade (*n* = 26)	15–16	77.92 ± 0.14	65.60 ± 3.71 (61.16 ± 3.64 /70.04 ± 3.80)	65.50 ±3.72	17.051	0.001	4.69	0.92	17.125	0.001	4.72	0.92
Anova	F	0.796	8.413	8.182	-	-	-	-	-	-	-	-
	g/l	3	3	3	-	-	-	-	-	-	-	-
	Sig.	0.498	0.00	0.00	-	-	-	-	-	-	-	-

^a^ 1st shows differences with the others.

**Table 4 ijerph-19-00020-t004:** Proposed furniture assignment according to the EU and Galician catalogues.

	Seat Size Galicia	Seat Size UE	Desk Size Galicia	Desk Size UE
Grade	Size	Freq.	%	Size	Freq.	%	Size	Freq.	%	Size	Freq.	%
1st grade (*n* = 32)	S36	10	31.3	S35	2	6.3	D60	21	65.6	D59	15	46.9
S40	21	65.6	S38	24	75.0	D66	10	31.3	D64	15	46.9
S44	1	3.1	S43	6	18.8	D72	1	3.1	D71	2	6.3
2nd grade (*n* = 32)	S36	3	9.4	S38	15	46.9	D60	6	18.8	D59	3	9.4
S40	22	68.8	S43	16	50.0	D66	26	81.3	D64	24	75.0
S44	6	18.8	S46	1	3.1				D71	5	15.6
S48	1	3.1									
3rd grade (*n* = 42)	S36	6	14.3	S35	4	9.5	D60	13	31.0	D59	5	11.9
S40	21	50.0	S38	16	38.1	D66	23	54.8	D64	24	57.1
S44	14	33.3	S43	18	42.9	D72	6	14.3	D71	13	31.0
S48	1	2.4	S46	4	9.5						
4th grade (*n* = 26)	S36	4	15.4	S35	2	7.7	D60	5	19.2	D59	3	11.5
S40	16	61.5	S38	11	42.3	D66	15	57.7	D64	16	61.5
S44	6	23.1	S43	12	46.2	D72	6	23.1	D71	7	26.9
			S46	1	3.8

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
