# Peer review of "Validation of Two Instruments for the Correct Allocation of School Furniture in Secondary Schools to Prevent Back Pain"

_ijerph, 2021, doi:10.3390/ijerph19010020_

Round 1
Reviewer 1 Report
The authors chose an objective on the mismatch between the anthropometric characteristics of students with the school furniture. Since the school furniture is same the anthropometric parameters will definitely not match because the people/students have different heights.
The objective of the study as stated by the authors was "Therefore, the objectives of this research are to analyze the degree of adjustment of
furniture in relation to the anthropometric characteristics of secondary schools students, check if the size system of the European reference is adjusted to the population under study and validate a system that allows teachers to reliably assign school furniture".
I am not sure if the authors have addressed the objective and if appropriate statistics were used to addess the objectives. A reviewer who is well qualified statistician should review this paper. Quality of writing also need to be improved. It needs a major revision.
Reviewer 2 Report
Does each progressive grade get a larger chair and larger desk?
Girls may grow initially faster and then boys grow faster and exceed girls usually in height. Any suggestions for girl/boy chairs.
Would modular chairs and desks be better?
How accurate is the the anthropometer?
Reviewer 3 Report
The purposes of this research are to analyze the degree of adjustment of furniture in relation to the anthropometric characteristics of secondary schools students, check if the size system of the European reference is adjusted to the population under study and validate a system that allows teachers to reliably assign school furniture. The manuscript provides information on the evident mismatch between the school furniture and the anthropometric measurements of the students, and the validation of the instrument created to determine the "ideal height" through the correlation with the gold-standard (anthropometer).
However, the second purpose set (to check if the size system of the European reference is adjusted to the population under study) has not been reflected in the results section.
Some aspects to improve the writing of the manuscript and presentation of the results are detailed below.
INTRODUCTION
The introduction presents a clear, well-structured wording that reflects the problem under study.
Line 93: "and validate a system that allows teachers to reliably assign school furniture". Where it says "reliable", I think it refers to "assigning school furniture in a valid/adequate way", given that the reliability you have studied is not that of the teachers using your new instrument, but of the measurer that have been part of the assessment group.
METHOD
In my opinion, the method is not written in a clear and orderly way. I believe that information is lacking to be able to replicate the study properly.
The explanation of the sample used is correct.
The explanation of the instruments should be in the same order as with the study objectives:
-"analyze the degree of adjustment of furniture in relation to the anthropometric characteristics of secondary schools students". What instruments do you use to test this objective?
-"check if the size system of the European reference is adjusted to the population under study". What is that reference system on which you base to make the verification
-"and validate a system that allows teachers to reliably assign school furniture". What is the new instrument and how is it made? If you want the teachers to use it, they will have to know how to get it.
When it comes to explaining the instruments you have used for the study, you should also explain what you have used them for, that is, what variables you have measured with those instruments.
You indicate that you have used an anthropometer (the gold-standard), but it does not refer to which variables you have measured with it, and suddenly in the results all the variables have appeared (Height, Weight, Popliteal Height, Elbow Height, Shoulder Height). You must explain how you have taken these measures and under what conditions (dressed, barefoot, in a group, alone, etc).
When you explain the new tool you are going to validate (The Ideal Seat Height Test and the Ideal Desk Height Test), you indicate that you designed it following the measures proposed by the European reference, what are those references?
Line 127: Peter lower leg meter, they should explain what it is.
Figure 1: very poor quality. You should improve the quality of the image and if you could include an image assessing a student (covered face), it would be ideal.
In section "2.3.3. Equations for the mismatch calculation" the initials "PH" appear for the first time. They should specify before or in the same section what they mean, in the same way as with EHS, and the rest of the acronyms.
Line 201: How do you classify the values obtained from the correlations? From what value is the correlation considered high or low?
RESULTS
Apparently the results are presented in the same order as the stated objectives. Although it includes too much information and results that have nothing to do with the objectives set, and other information that has more space in the "Discussion" section, such as: line 265.
Results related to objective 2 are also not included.
Tables 2 and 3 become difficult to read, the name "Ideal height / range 1" could be replaced by "height with anthropometer", as well as "ideal height 2" by "height with ISHT or IDHT". Why in the measurements with an anthropometer you put a range and in the measurements with your new instrument not?
Image quality 2 should be improved.
DISCUSSION
The discussion presents a good writing, but depending on the changes that are made in the results, it may have to be revised.
I think that one of the future lines would be to test the reliability of your tool with teachers, since in this study the tool has been used by expert and trained researchers. A tool can be valid, but it also needs to be reliable.
Reviewer 4 Report
Dear authors,
I consider that it is a work that can be published in the journal due to its quality, but there is a work by the same authors, aimed at analyzing furniture in the primary stage, published in the journal: Rev Esp Salud Pública. 2021; Vol. 95: September 15th which is not cited in the present manuscript.
As they are the same authors and with the same subject of analysis of school furniture and that differ in the stage to which they are addressed (primary education and secondary education), the work should be cited throughout the article since many issues raised are very similar.
The full reference is:
Paramés González, Adrián & Santiago, Alfonso & Santiago, Jesús & Prieto Lage, Iván. (2021). Back pain prevention through the correct allocation of school furniture: validation of two instruments. Revista Española de Salud Pública, 95.
Round 2
Reviewer 3 Report
Congratulations on the great work you have done to improve the manuscript.
Here are some suggestions that you may or may not accept.
In the formula used, some acronyms are wrong. Instead of SHH, it would be SHS?.
I'm glad they changed the method titles, but I would make them shorter and more concise.
Point 2.2 would be titled "Procedure".
The 2.2.1. "Anthropometric characteristics and furniture".
Line 135, the formulas you describe are to determine "the ideal seat and desk heights", and then with the real measurements of the furniture you see if there is a match or mismatch. Since the important thing is "the ideal seat and desk heights" I consider that you should emphasize it well in this paragraph, so that the objective of the formulas that you put is clear. In the case that the formulas are not to determine "the ideal seat and desk heights", the sentence in line 152 "Once the ideal seat and desk heights were determined for each student", would not make sense.
Point 2.2.2. "Size system of the EU and Galician catalog and its adjustment to the study population", too long. Proposal "Size system of the EU and Galician catalog".
Line 155, could you explain what the EU catalog is based on to establish the 8 sizes? If it is not by school year as they say, how are the sizes distributed?
Idem with the Galician catalog.
Point 2.2.3. "Measurement tools used to validate (ISHT and IDHT)". Proposal: "Ideal Seat Height Test (ISHT) and Ideal Desk Height Test (IDHT)". In the text you already explain that they are tools that you are going to validate.
Line 168. Thank you for including the link, so it will be of great help to readers who wish to use it.
Figure 2. Thanks for adding the images, it is much clearer. Although they have not named the image in the text. Include it.
Point 2.3. "Procedure". Proposal: 2.2.4. "Data collection", and there they explain both the training and the previous reliability study.
Author Response
"Please see the attachment."
